# DATAS$^3$:
# DATASET SUBSET SELECTION FOR SPECIALIZATION

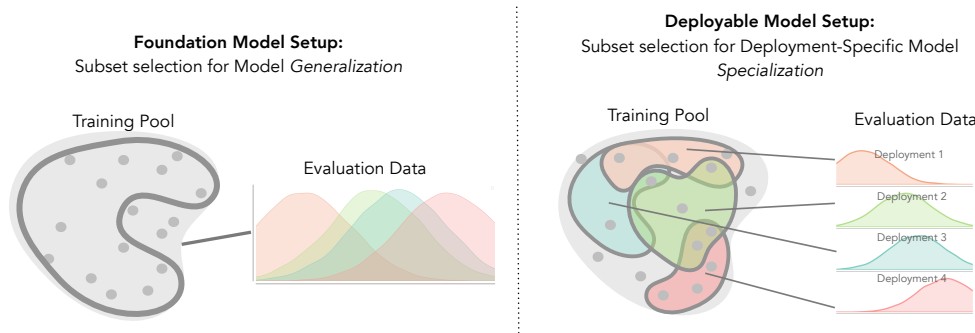

Figure 1: Foundation model training aims for broad generalization, by using all data available, usually from massive internet-scale datasets. In practice, we find these models are often suboptimal for specific deployments, which may exhibit different distributions over categories or data characteristics from the general training data pool. Dataset subset selection for specialization seeks to identify model training subsets closely aligned with the target deployment, achieving superior performance under the given distribution and attribute shifts.

## ABSTRACT

In many real-world machine learning applications (e.g. detecting broken bones in x-rays or species in camera traps), models need to perform well on specific deployments (e.g. a specific hospital or national park) rather than the domain broadly. However, deployments often have imbalanced, unique data distributions. Discrepancies between training and deployment distributions lead to suboptimal performance, highlighting the need to curate training data for *specialized models for specific deployment needs*. We formalize **dataset subset selection for specialization (DS3)**: given a training set drawn from a general distribution and a (potentially unlabeled) query set drawn from a deployment-specific distribution, the goal is to select a subset of the training data that optimizes deployment performance.

We introduce DATAS$^3$; the first dataset and benchmark designed specifically for the DS3 problem. DATAS$^3$ encompasses five *real-world* application domains, each with a set of distinct deployments to specialize in. We conduct a comprehensive study evaluating different state-of-the-art data curation algorithms and find that methods trained on general distributions consistently fail to perform optimally on deployment tasks. Additionally, we demonstrate the existence of expert-curated (deployment-specific) subsets that outperform training on all available data by up to 51.3%. Our benchmark highlights the critical role of tailored dataset curation in enhancing performance and training efficiency on deployment-specific distributions, which we posit will only become more important as global, public datasets become available across domains and ML models are deployed in the real world.

## 1 BACKGROUND AND MOTIVATION

Machine learning models are typically trained on large datasets with the assumption that the training distribution closely matches the distribution of the deployment where the model will be applied. However, in real-world applications, deployment data distributions often diverge from general and/or

global training set distributions (Shen et al., 2024; Taori et al., 2020). Selecting relevant data subsets aligned with specific deployments is crucial to maximize field performance. The problem of *data subset selection for specialization* (DS3) is thus critical: given all available training data for a domain and a small (usually unlabeled) query set that represents the desired deployment, the goal is to identify a subset of the training data, such that training the ML model on this subset maximizes performance on the deployment distribution.

**Real world example.** Consider a wildlife ecologist who aims to build a classifier to detect the presence of invasive species in camera trap images collected at the Channel Islands. Existing labeled training data in this context is limited, thus training a classifier from scratch is likely to be unsuccessful. A common approach is to finetune a general pre-trained model (such as ViT or CLIP) on all *relevant* camera trap data. But *what does "relevant data" mean?* Would using similar species data from other camera trap locations (perhaps on the mainland) improve performance, or introduce noise? What about including data from non-similar species at that location? While adding data to a training set can sometimes improve performance, it can also decrease individual subgroup performance in a biased way (Compton et al., 2023) and introduce spurious correlations that can enable models to learn potentially dangerous "shortcuts," resulting in biased predictions, shown across various domains (Geirhos et al., 2020; Badgeley et al., 2018; Wang et al., 2021; Beery et al., 2022a).

**Our contributions.** Our key contributions are the following:

(i) We are the first to identify and formalize the challenge of sub-selecting training data to specialize models to new deployments (dataset subset selection for specialization).

(ii) We propose $\textsc{Datas}^3$: A novel benchmark that enables the AI community to investigate and make progress on DS3. $\textsc{Datas}^3$ reformulates, adapts, and adds to five diverse datasets, each from a different application domain. We worked directly with domain experts throughout the curation and reformulation process to ensure that $\textsc{Datas}^3$ accurately reflects **(1)** real-world dataset distribution challenges that require model specialization (i.e., covariate shifts, subpopulation shifts, and long-tailed distributions), and **(2)** evaluation settings (test splits) representative of real-world deployment scenarios in each domain.

(iii) We show that a well-curated subset can consistently outperform models trained on the entire dataset for each deployment.

(iv) We also conduct an extensive experimental study comparing current SOTA subset selection methods on $\textsc{Datas}^3$. After training a suite of baselines, our results clearly show that current subset selection methods fail on DS3, highlighting the need future research to solve the DS3 problem on $\textsc{Datas}^3$.

(v) We release a codebase, python package, and public leaderboard for submission to the benchmark, available at `datas3-benchmark.github.io`

## 2 PROBLEM STATEMENT

**DS3 problem formulation.** Let $X$ be a pool of data points, $T \subset X$ be a given *training set* drawn from a training (pool) distribution $P_T$ over $X$, and let $Q \subset X$ be a *query set* drawn from the desired **deployment-specific distribution** $P_Q$ over $X$. Given a model $\theta$, the objective of **dataset subset selection for specialization (DS3)**, is to design an algorithm `SubsetSelection-ALG`, which takes $T$ (the training set) and $Q$ (the deployment representative query set) as input, and outputs a subset $S^* \subset T$ that minimizes the expected loss of $\theta$ trained on $S^*$ over the desired deployment-specific distribution $P_Q$. More formally:

$$S^* = \underset{S \subset T}{\arg\min} \, \mathbb{E}_{q \sim P_Q} \left[ \mathcal{L}(\theta(S), q) \right], \tag{1}$$

where $\theta(S)$ denotes the model trained on the subset $S \subset T$, and $\mathcal{L}(\theta(S), q)$ is the loss function evaluated on a single point $q$ sampled from $P_Q$ and the trained model $\theta(S)$. The term $\mathbb{E}_{q \sim P_Q}$ denotes the expected value over the distribution $P_Q$. Hence, the algorithm `SubsetSelection-ALG` outputs $S^*$, the subset of $T$ that minimizes the expected loss of the entire desired deployment distribution $P_Q$. Notably, `SubsetSelection-ALG` can only access the desired deployment-specific distribution via the query set $Q$. Unlike complementary lines of work such as active domain

adaptation (ADA), which assumes real-time compute and focuses on actively/iteratively selecting and then collecting labels for data within the deployment during the specialization process, DS3 selects data in a single-shot approach prior to specialization on an already available pool of data (potentially for use in resource-constrained applications).

**Is the query set labeled?** This formalization can be divided into two cases. In the first, the query set $Q$ is annotated with a set of labels: $Q$ is a set of $m > 0$ pairs $Q = \{(q_1, y_1), \cdots, (q_m, y_m)\}$, where for every $i \in [m]$, $q_i$ is the $i$th feature vector describing the $i$th input, and $y_i$ is it corresponding label/annotation. In this case the algorithm `SubsetSelection-ALG` has access to the set of labels $\{y_1, \cdots, y_n\}$. In the second scenario, no labels are provided for $Q$, meaning that the `SubsetSelection-ALG` does not have access to the set $\{y_1, \cdots, y_n\}$ and consequently $Q = \{q_1, \cdots, q_m\}$. In a real-world example, $Q$ can be thought of as the data collected from a deployment thus far, enabling additional selection from a larger database (the training pool). Annotating $Q$ for any specific deployment is quite expensive, requiring time, money, and expertise, so progress on methods without query labels would helpful for real-world applications.

**Is `SubsetSelection-ALG` model agnostic?** Similarly, this formalization can be approached in two different ways: one where the computation of $S^*$ depends on a given specific model $\theta$, i.e., `SubsetSelection-ALG` is model dependent, and has access to the model $\theta$ we wish to train on. Ideally, a well-performing, robust method should work well for multiple models, and will be more generalizable than a model-dependent algorithm. We test several different models on our benchmark for various `SubsetSelection-ALG` baselines to test this.

**Should `SubsetSelection-ALG` be sample efficient?** The goal of our benchmark is to specialize on a desired deployment distribution. Unlike standard subset selection, where subset size is often a primary concern, our focus is on selecting subsets based on relevance based on a particular deployment that yield highest performance evaluated on that deployment. Smaller subsets offer many advantages, such as training efficiency, lower memory/storage, etc; we analyze these tradeoffs in Appendix C.

## 3 RELATED WORK

It has become increasingly clear that data work is equally important to architecture design for increased model performance (Compton et al., 2023). Data curation for better quality training pools has been identified as an important line of research within this field. Many methods have been proposed for data curation and subset selection – we provide a comprehensive overview of these methods in Appendix B. Current benchmarks for data curation include Gadre et al. (2024), Mazumder et al. (2023) and Feuer et al. (2024). However, these benchmarks focus on data curation for a single higher quality training pool meant for better performance across many different downstream tasks, in contrast to specialization for a particular deployment. Additionally, data selection methods (Killamsetty et al., 2021b; Tukan et al., 2023) are often evaluated on standard CIFAR10/100 (Krizhevsky et al., 2009) or ImageNet (Deng et al., 2009b) datasets, where test and validation sets have similar distribution to their training sets. No existing benchmarks focus on the DS3 challenge. DATAS$^3$ is the first benchmark specifically designed to evaluate subset selection methods for *deployment-specific specialization*, rather than generalization, where the training and testing data exhibit distributional shifts representative of real-world deployment challenges (Figure 1).

## 4 THE DATAS$^3$ BENCHMARK

**Datasets.** We describe each DATAS$^3$ dataset. Our benchmark includes five datasets, each capturing a unique and diverse application of ML: AutoArborist for tree classification (Beery et al., 2022b), iWildCam for camera trap species identification (Beery et al., 2021), GeoDE for diverse object classification (Ramaswamy et al., 2023), NuScenes for autonomous driving footage steering regression (Caesar et al., 2020), and FishDetection for underwater video fish detection (Dawkins et al., 2017). To make the DATAS$^3$ datasets usable, we have made considerable changes to them to better highlight deployment challenges, augment with additional data, or preprocess the data for use with standard ML pipelines. For each dataset, we provide a proof-of-concept "oracle" / knowledge-driven subset that demonstrates the usefulness of subset selection, with improvements over using training on all data. These subsets were created using information that benchmark users are not provided (e.g. metadata, GPS location, region, etc). Additional details about each dataset and can be found in Apdx. E.

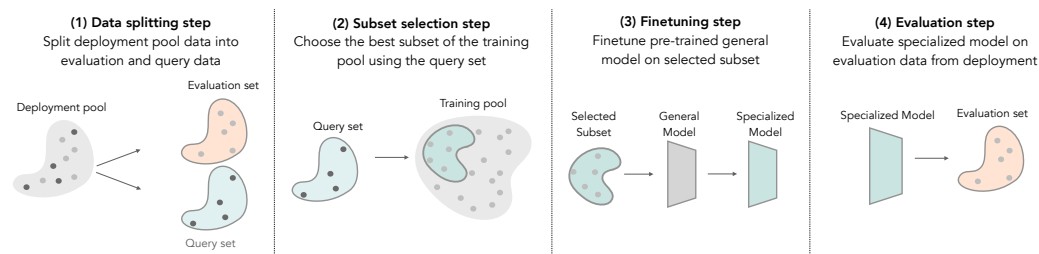

Figure 2: $\mathrm{DATAS^3}$ benchmark process, involving dataset splitting, subset selection, model specialization/finetuning, and then evaluation.

## 4.1 iWildCam

**Motivation:** Animal populations have declined by $68\%$ on average since 1970 (Staub, 2020). To monitor this biodiversity loss, ecologists deploy camera traps—motion-activated cameras placed in the wild (Wearn & Glover-Kapfer, 2017)—and process the data with machine learning models (Norouzzadeh et al., 2019; Beery et al., 2019). However, variations in illumination, camera angle, background, vegetation, color, and animal frequencies across different locations cause these models to generalize poorly to new deployments. To specialize models for specific locations, selecting appropriate data subsets for deployment-specific (in this case location) specialization becomes essential.

**Problem Setting & Data:** To study this problem, we use the iWildCam 2020 dataset, comprising of $203,029$ images from $323$ different camera traps spread across multiple countries in different parts of the world. The task is multi-class species classification from $182$ different animal species. Performance is measured by overall classification accuracy for species identification. The original camera trap data comes from the Wildlife Conservation Society (link).

**Deployments:** Our deployments were defined to be split across camera trap locations to simulate the common scenario of researchers setting up new cameras within a region, with poor model generalization on the new cameras (Wearn & Glover-Kapfer, 2017). Our train/test split was done randomly across the 200 locations, with the five downstream test tasks created by clustering by the latitude and longitude of camera GPS location in 4 deployments: (1) Central America, (2) Eastern Africa, (3) Southern Africa, and (4) Southeast Asia. Similar to most other camera trap datasets, iWildCam has significant long-tailed label distributions, with variation in species and backgrounds between locations, as can be seen in Figure 3.

**Knowledge-driven Subset:** These subsets were created by only choosing training data from camera locations that are within 100km of the camera locations in the deployments (the relevant geographical area) and eliminating irrelevant classes that are not present in the deployment.

## 4.2 GeoDE

**Motivation:** Object classification datasets are often constructed by scraping images from the web but contain geographical biases (Shankar et al., 2017). Instead of scraping images from the web, GeoDE (Ramaswamy et al., 2023) crowdsources a dataset that is roughly balanced across 40 different objects and six world regions, showing that common objects (stoves, bicycles, etc), vary in appearance across the world. Accordingly, specializing models to different regions becomes useful when the objects have strong covariate shift.

**Problem setting & Data:** GeoDE is a diverse dataset of 61,490 images comprising 40 different objects collected from 6 world regions (Africa, Americas, East Asia, Europe, Southeast Asia, West Asia). The associated task is multiclass classification, where the goal is to predict the object depicted in each image.

**Deployments:** We propose 4 different deployments: (1) objects in Indonesia, (2) objects in Nigeria, (3) indoor objects, and (4) outdoor objects, as shown in Figure 3. Nigeria and Indonesia were selected as the two countries with the poorest performance, and the indoor/outdoor deployment tasks were

Figure 3: The five datasets in our benchmark: iWildCam, GeoDE, AutoArborist, FishDetection, and NuScenes each have real-world applications in deployment. In iWildCam, GeoDE, and AutoArborist, we show the class distributions of each deployment; in FishDetection, the number of detections per image is shown, and in NuScenes environment features. These diagrams show that each dataset has unique challenges in the deployments that lead to a need for model specialization, including long-tailedness (AutoArborist, iWildCam), covariate shift (all), subpopulation shifts (GeoDE, FishDetection), and more. These axes of variation are described in depth in Section 4 and further in Apdx E.

selected for enabling model specialization. The training dataset includes images from all countries, and the test data contains only images from Nigeria and Indonesia.

**Knowledge-driven Subset:** These subsets were generated by selecting data from the relevant countries/categories in the training data, ie. only selecting African subcontient data for the Nigeria deployment, Asian subcontinent data for the Indonesia deployment, and indoor/outdoor objects within the training pool for these deployments.

### 4.3 AUTOARBORIST

**Motivation:** Ecological imagery for environmental monitoring, such as automated tree classification, provides policymakers with critical, data-driven insights to support climate adaptation, urban planning, and more (Brandt et al., 2016). This task is associated with fundamental challenges such as noisy labels, non-iid data, fine-grained and long-tailed class distribution, and geospatial distribution shift. These challenges lead to a need for specialization of models where general-purpose models fail.

**Problem Setting & Data:** The AutoArborist dataset is a multi-view, fine-grained visual tree categorization dataset containing street-level images of over 1 million public zone trees from 300 genus-level categories across 23 major cities in the US and Canada.

**Deployments:** Deployments in AutoArborist correspond to the development models for use by individual cities. The deployment cities of (1) Surrey with 66 distinct tree genus classes, (2) Calgary with 30 classes, (3) Los Angeles with 175 classes, and (4) Washington DC with 67 classes were chosen due to their diverse climates, species distributions, and urban structures, as seen in Figure 3. Surrey and Calgary were treated as our in-distribution (ID) deployments, with some of these cities data in the training pool. Washington DC and LA were the out-of-distribution deployments, with no city data in the training pool.

**Knowledge-driven Subset:** We used the relevant data from Surrey and Calgary in the training pool for these ID deployments. Accordingly, we used data from San Francisco and San Jose for Los Angeles and Charlottesville, Pittsburgh, and New York for Washington DC. Label distribution shift and covariate shift are visualized in Figure 10 and 9, respectively.

### 4.4 FISHDETECTION

**Motivation:** Climate change, pollution, and overfishing continue to threaten marine biodiversity across the globe (United Nations, 2023; Di Lorenzo et al., 2022). Marine imagery is an increasingly common resource to monitor fish stocks and biodiversity. However, ML methods are difficult to apply across various environmental settings due to differences in lighting, turbidity, species, vegetation, camera sensors, etc. (Borremans et al., 2024; Jerlov, 1976; Akkaynak & Treibitz, 2019), creating a need for specialized models.

**Problem Setting & Data:** We use the public VIAME FishTrack23 dataset (Dawkins et al., 2017) consisting of 854,078 images across various environmental settings, ranging from freshwater rivers to deeper benthos. Specifically, the task is to predict bounding box localizations around every fish present in each image. Performance is measured by mAP across various IoU thresholds. Most of the images across all datasets are taken from video streams, and can be grouped as such, that were deployed primarily on camera traps, both baited and unbaited.

**Deployments:** Deployments are split according to the subsets of the VIAME dataset, which roughly correspond to geographic regions. Train, test and subset splits are either taken as provided or randomly sampled frames from each dataset, roughly corresponding to: (1) freshwater Pacific Northwest lakes; (2) Pacific Ocean; (3) East Atlantic Ocean; and (4) Gulf of Mexico.

**Knowledge-driven Subset:** For each deployment, we use the subset in the relevant geographical area (e.g., images from Gulf of Mexico for the Gulf of Mexico deployment).

### 4.5 NUSCENES

**Motivation:** End-to-end autonomous driving systems streamline vehicle control by directly mapping sensory inputs, such as images, to control outputs like steering angles (Wang et al., 2024). Adapting these systems to specialize in particular streets or environments is made easier as a single model

encompasses the full system. Thus, training this model to specialize in a specific environment brings advantages, capturing detailed local road layouts, traffic patterns, area-specific obstacles, and more.

**Problem Setting & Data:** We explore vision-based control for self-driving across diverse environments (e.g., different city areas) and driving scenarios (e.g., pedestrians crossing, construction zones), formulated as a regression task. This dataset includes $88,461$ images from the NuScenes dataset, subsampled from the image sweeps at a rate of 2. The images were captured from a video stream recorded while driving a car. Each image is paired with a steering angle control from the CAN bus, synchronized with the sensor timestamps of both the camera and CAN bus data. The model's goal is to predict a single scalar value representing the car's steering angle. Performance is evaluated in an open-loop manner using metrics like mean squared error.

**Deployments:** Deployments are organized by the geographic locations where the data was collected, including (1) Boston Seaport, (2) Singapore Holland Village, (3) Singapore One-North, and (4) Singapore Queenstown. While all tasks are based on expert demonstrations of driving and general driving behaviors, each location presents varying environmental features—such as vegetation, road types, roadside infrastructure, and weather—as well as differences in driving style and road regulations. Train/test splits are randomly sampled within each deployment.

**Knowledge-driven subset:** Since this training pool is a combination of the four deployment locations, we simply use the relevant location's data as the training subset. For example, we use the subset of the training pool with Boston Seaport data for the Boston Seaport deployment.

### 4.6 BENCHMARK PIPELINE

To compete on our benchmark, models must select relevant data from the training pool and then finetune models on that relevant data. Explicitly, **(i)** given a small query set representing the deployment data (we consider both labeled and unlabeled query sets), curate a subset of data from the training for a specific deployment, **(ii)** finetune/train a fixed model on the chosen subset from the training pool and **(iii)** evaluate on the deployment (test) set (Figure 2).

For each dataset, we fix the training procedure for all subsets, fixing model architecture, optimizers, and loss functions. We match the label distribution of the query set to the deployment/test set as closely as possible using stratified sampling, but from each class of the training pool, we sample uniformly at random. We run a small hyperparameter sweep for each training subset across batch sizes $\{32, 64, 128\}$ and learning rates $\{0.01, 0.001, 0.0001\}$ for each deployment. For all classification/regression datasets, we use ResNet50 for full-finetuning (He et al., 2015) (table 1) and a ViT for LoRA finetuning (Apdx Table 3), as well as a ViT (Dosovitskiy et al., 2020) for linear probes (Apdx Table 2). For the detection dataset, we use a YOLOv8n model, using default parameters, though we subsample images to 640p. Full details are in Apdx D.

### 4.7 METRICS

Participants are evaluated across 12 deployments from five datasets, as outlined in Section 4. For the classification task datasets of GeoDE, AutoArborist, and iWildCam, we report accuracy for each deployment, for the regression task dataset NuScenes, we report mean squared error, and for the detection task FishDetection, we report mAP50. For each deployment, we evaluate participants of the benchmark on overall accuracy of training subset; we also report subset size – while the less data used the better, we mainly focus on optimal performance, in line with the DS3 formulation.

## 5 BASELINES

We compare performance of dataset subset selection algorithms across our benchmark, across different scenarios: (a) access to an unlabeled query set, and (b) access to a labeled query set. We also curate a third category, (c), which leverages domain expertise to generate expert-selected subsets, in order to demonstrate the existence of better-than-all subsets for these deployments.

**Non-subset comparisons:**
*No filtering:* Performance of a model trained on the entire training pool, without any filtering.

*Query Sets:* As a comparison, we also include performance of a model trained directly on the labeled query set for each deployment. Note that this would require access to query labels, which are not always available. When labels are available, performance of models trained on the small query sets are often poor, hence the value of learning from larger-scale general-pool data. As a logistical point, none of the baselines we show in our results train on query set data.

**Expert-Driven Subsets:** We contribute curated, "expert knowledge" subsets using domain knowledge and/or metadata. We find these knowledge-guided subsets often outperform using all samples in the training pool (no filtering). The creation of these subsets is described per-dataset in section 4.

**Unlabeled-query baselines:**

*Image-alignment (Image-Align):* We take the cosine similarity between the training and query embedding space, using examples that exceed a threshold for at least $x$ samples, where $x$ is a hyperparameter chosen from $\{1,10,100\}$.

*Nearest neighbors features (Near-Nbors):* To better align our method with the downstream deployment, we explore using examples whose embedding space overlaps with the query set of data. To do so, we cluster image embeddings extracted by an OpenAI ViT model for each image into 1000 clusters using Faiss (Johnson et al., 2019). Then, we find the nearest neighbor clusters for every query set example and keep the training cluster closest to each query set cluster. This method was inspired by the similar DataComp baseline (Gadre et al., 2024).

**Labeled-query baselines:**

*CLIP score filtering (CLIP-score):* We also experiment with CLIP score filtering, using examples that exceed a threshold for cosine similarity between CLIP image and text similarity. Text for each image was created with manual captioning (e.g. for iWildCam, *"This is a camera trap image of a lion taken at time 10-2-2016 at 04:26:13 in Nigeria"*). We select the subset that exceeds a threshold of CLIP-score similarity, with the threshold calculated for subsets that make up $25\%, 50\%, 75\%$, and $90\%$ of the dataset.

*Matching relative frequency (Match-Dist):* We explore having access to the relative frequency of each label in the downstream deployment. For example, a domain expert at a national park might know the relative frequency of species (deployment-specific domain knowledge). We create subsets by sampling $25\%, 50\%, 75\%$, and $90\%$ of the training pool to match the label distribution of the deployment.

*Matching labels (Match-Label):* Similarly, a domain expert may know the classes present in the downstream deployment. For example, a domain expert at a national park might know the species present (deployment-specific knowledge) that we can utilize for subset selection. For these subsets, we simply remove the classes present in the training pool that are not present in the testing pool.

## 6 RESULTS

**Well chosen subsets outperform training on all data**. The knowledge-driven subsets in Table 1 show that deployment-specific well-chosen subsets of the data can significantly outperform models trained on all the data, with improvements in deployment accuracy up to 3.6% for GeoDE, 11.9% for iWildCam, 51.3% for AutoArborist, a 0.03 reduction in MSE for NuScenes, and 0.13 increase in mAP50 for FishDetection. Even when the knowledge-driven subsets underperform all training data, as in NuScenes Deployment 2, there exist subsets from other baselines that outperform using all the data. In Appendix E, we provide a additional breakdown of the key factors that contributed to performance gain on these knowledge-driven subsets.

**Training on more data has diminishing returns.** For all deployments, we see that we can achieve near-optimal performance with subsets of the data. The knowledge-driven subsets are significantly smaller than the total training data size, with the average percentage of the total training pool used being 4% for GeoDE, 11% for iWildCam, 8% for AutoArborist, 10% for NuScenes, and 20% for FishDetection. Appendix C shows that even 25% of the data can perform near-optimally in some cases, with little performance loss with 50% of the data on the algorithmic baselines. Overall, these results demonstrate that greater efficiency for training specialized ML models is possible, potentially reducing computational and data storage burdens in deployable settings.

| Dataset | Metric | Deploy # | Non subset | | Knowledge-driven | Unlabeled query set | | Labeled query set | | |
|---|---|---|---|---|---|---|---|---|---|---|
| | | | Query-set | All-data | | Image-Align | Near-Nbors | CLIP-score | Match-Label | Match-Dist |
| GeoDE | Acc (#) | Deploy 1 | 0.87 (500) | 0.89 (53k) | **0.92 (2.9k)** | 0.88 (26k) | 0.88 (48k) | 0.89 (40k) | 0.88 (53k) | 0.89 (48k) |
| | | Deploy 2 | 0.45 (500) | 0.89 (53k) | **0.91 (2.6k)** | 0.90 (26k) | 0.89 (48k) | 0.90 (40k) | 0.90 (53k) | 0.88 (27k) |
| | | Deploy 3 | 0.95 (500) | 0.82 (53k) | **0.85 (1.4k)** | 0.85 (24k) | 0.76 (48k) | 0.84 (40k) | 0.83 (1.4k) | 0.88 (48k) |
| | | Deploy 4 | 0.83 (500) | 0.83 (53k) | **0.85 (2.6k)** | 0.79 (24k) | 0.78 (48k) | 0.83 (40k) | 0.84 (2.6k) | 0.83 (13k) |
| iWildCam | Acc (#) | Deploy 1 | 0.70 (301) | 0.66 (130k) | 0.65 (8.5k) | 0.56 (36k) | 0.50 (117k) | 0.50 (97k) | 0.74 (8.1k) | **0.74 (117k)** |
| | | Deploy 2 | 0.78 (302) | 0.34 (130k) | 0.35 (9.2k) | 0.44 (45k) | 0.47 (98k) | 0.46 (97k) | 0.35 (55k) | **0.49 (65k)** |
| | | Deploy 3 | 0.44 (301) | 0.72 (130k) | 0.75 (19k) | 0.54 (24k) | 0.45 (98k) | 0.42 (97k) | 0.72 (60k) | **0.75 (117k)** |
| | | Deploy 4 | 0.46 (309) | 0.66 (130k) | 0.67 (21k) | 0.60 (22k) | 0.60 (33k) | 0.29 (97k) | 0.69 (57k) | **0.74 (33k)** |
| AutoArborist | Acc (#) | Deploy 1 | 0.16 (1.5k) | 0.35 (781k) | **0.86 (70k)** | 0.38 (44k) | 0.39 (391k) | 0.38 (47k) | 0.67 (368k) | 0.74 (703k) |
| | | Deploy 2 | 0.20 (1.5k) | 0.48 (781k) | **0.86 (123k)** | 0.11 (49k) | 0.14 (703k) | 0.14 (47k) | 0.65 (532k) | 0.56 (391k) |
| | | Deploy 3 | 0.12 (1.5k) | 0.16 (781k) | **0.38 (35k)** | 0.16 (46k) | 0.10 (703k) | 0.17 (47k) | 0.16 (534k) | 0.23 (703k) |
| | | Deploy 4 | 0.12 (1.5k) | 0.14 (781k) | **0.39 (26k)** | 0.10 (48k) | 0.11 (391k) | 0.11 (47k) | 0.10 (527k) | 0.23 (195k) |
| NuScene | MSE (#) | Deploy 1 | 0.063 (6.0k) | 0.050 (100k) | **0.029 (20k)** | 0.040 (35k) | 0.040 (90k) | 0.073 (31k) | - | - |
| | | Deploy 2 | 0.070 (1.0k) | 0.021 (100k) | 0.049 (4.6k) | 0.15 (17k) | 0.042 (90k) | **0.032 (31k)** | - | - |
| | | Deploy 3 | 0.089 (2.7k) | 0.068 (100k) | **0.038 (10k)** | 0.049 (28k) | 0.13 (90k) | 0.071 (31k) | - | - |
| | | Deploy 4 | 0.12 (1.9k) | 0.048 (100k) | **0.039 (7.0k)** | 0.086 (26k) | 0.39 (90k) | 0.050 (31k) | - | - |
| FishDetection | mAP50 (#) | Deploy 1 | 0.22 (500) | 0.68 (841k) | **0.69 (179k)** | 0.50 (630k) | 0.60 (103k) | - | - | - |
| | | Deploy 2 | 0.26 (600) | 0.32 (841k) | **0.45 (152k)** | 0.31 (630k) | 0.40 (120k) | - | - | - |
| | | Deploy 3 | 0.13 (541) | 0.32 (841k) | **0.39 (6.0k)** | 0.28 (630k) | 0.23 (204k) | - | - | - |
| | | Deploy 4 | 0.079 (519) | 0.59 (841k) | **0.60 (320k)** | 0.54 (630k) | 0.39 (45k) | - | - | - |

Table 1: Best-performing subsets across hyperparameters for baseline methods across all datasets and deployments (abbreviated as Deploy) for YOLOv8 full-finetuning for FishDetection and ResNet50 full-finetuning for the rest. Accuracy is reported for the classification tasks of GeoDE, iWildCam, and Auto Arborist, mAP50 for FishDetection (greater is better), and MSE for NuScenes (smaller is better). We include subset size in parentheses. We include results for **ViT LoRA finetuning and ViT linear probes** in Appendix C in Table 3 and Table 2, which display similar trends. Match-Dist and Match-Label are not applicable for NuScenes, as it is a regression task and does not have clear classes/labels for these methods. FishDetection only uses the unlabeled query set, as the ground truth is bounding boxes, rather than labels themselves. Baselines are distinguished from one another by their access to information, with each baseline having access to expert knowledge, or a labeled/unlabeled query set. We do not report the random baseline in this table, but demonstrate results in Appendix C as it mainly refers to subset size. For each deployment, there exists a subset that outperforms training on all data, indicated in bold.

**Methods without access to supervision perform poorly.** While the knowledge-driven subsets in Table 1 demonstrate that a well-chosen subset *does exist* for all deployments, finding this subset without extra knowledge is still an open problem. Some of our baselines require access to query labels, this requirement can in many cases be unrealistic in the deployable ML setting (labels can be expensive or difficult to collect). The two unsupervised baselines, the nearest neighbors and image alignment methods, do not perform optimally on the deployments, often underperforming using all the training data. Our benchmark opens up the line of research for potential unsupervised methods for this data subset selection process.

# 7 DISCUSSION AND CONCLUSIONS

DATAS[3] is the first benchmark to promote the development of dataset subset selection methods capable of specialization to diverse real-world deployments. The benchmark is both open-source and easy to use, lowering the barrier to entry for this new important problem.

**DATAS[3] highlights open challenges for the research community.** In our experimental study, we show that there is no winning baseline that performs well across multiple domains/datasets. Additionally, while some methods perform well when given access to labeled query sets, no methods perform well in the unsupervised setting. Finally, some datasets are more challenging than others– methods may need to specifically target different types of distribution shifts.

**DATAS[3] has value beyond subset selection.** In addition to the DS3 problem, DATAS[3] can be used as a testbed for various other complementary lines of work, such as domain adaptation, active learning, coreset selection, and more. We highlight these relevant methods in Appendix B.

**Extensions to other domains.** Model specialization for deployments isn't limited to the domains we include. We are open to expanding this benchmark to capture more scientific domains, and welcome further dataset contributions from the broader ML and scientific research community.

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
