## A CODEBASE AND DATA

We make all dataset splits and scripts to download the data and run baseline experiments open-source, available at `https://anonymous.4open.science/r/data-codebase-BD2D/`.

## B ADDITIONAL RELATED WORK

Traditional data subset selection approaches can be split into two main categories: 1) Data filtering/cleaning, which focuses on refining the dataset to enhance its quality (Zhang et al., 2022; Raffel et al., 2020), and 2) Coresets for dataset subset selection, aimed at reducing training time by a computing a subset that effectively represents the larger training dataset (Killamsetty et al., 2021b; Tukan et al., 2023).

**Data filtering for better learning.**  Data pruning is widely used in NLP to clean noisy datasets (Anonymous, 2023), often employing filtering and heuristics (Bane et al., 2022). Common methods include excluding texts with blocklisted words (Raffel et al., 2020), removing duplicates (Zhang et al., 2022), filtering out non-English texts (Raffel et al., 2020; Rae et al., 2022), and discarding short sentences (Raffel et al., 2020; Rae et al., 2022). Perplexity-based filtering removes high-perplexity samples considered unnatural and detrimental to performance (Muennighoff et al., 2023; Wenzek et al., 2020; Laurençon et al., 2023). Although simple filtering can enhance language models (Penedo et al., 2023; Raffel et al., 2020), their effectiveness varies, and some studies report no benefits (Black et al., 2022; Biderman et al., 2023), possibly due to their simplicity. (Zhou et al., 2024) showed that manually selecting a small subset satisfying quality and diversity improves alignment performance.

**For vision tasks,** a smaller number of methods have been suggested for data filtering (Sorscher et al., 2023) to obtain better trainable subsets (Siddiqui et al., 2022) through the use of model signals (Mindermann et al., 2022).

**Coresets for efficient learning.** Subset selection (hitherto referred to as coresets) is common for vision tasks. The goal is to compute a small subset from the training dataset, that approximates training on the full dataset, thus boosting the training process (Braverman et al., 2016; Maalouf et al., 2022). Coresets proved to be useful in many applications such as regression (Dasgupta et al., 2008; Chhaya et al., 2020; Tolochinsky et al., 2022; Meyer et al., 2022; Maalouf et al., 2019), clustering (Har-Peled & Mazumdar, 2004; Chen, 2009; Huang & Vishnoi, 2020; Jubran et al., 2020; Cohen-Addad et al., 2022), low-rank approximation (Cohen et al., 2017; Braverman et al., 2020; Maalouf et al., 2021), support vector machines (SVMs) (Clarkson, 2010; Tukan et al., 2021; Maalouf et al., 2022), and for compressing neural networks (Baykal et al., 2022; Liebenwein et al., 2019; Tukan et al., 2022). For boosting the training of neural networks, (Coleman et al., 2019) used proxy functions to select subsets of training data approximating the training process. Later (Mirzasoleiman et al., 2020a;b) developed algorithms to estimate the full gradient of the deep neural network on the training data and then further refined by (Killamsetty et al., 2021a;b; Paul et al., 2021; Wang et al., 2020). Other methods require a neural network forward pass to get embeddings (Sener & Savarese, 2018; Sorscher et al., 2022; Killamsetty et al., 2021c). All these methods assume the training data well represents the test (deployment) data, as the case in known diverse, high-quality vision benchmarks (CIFAR10 and ImageNet). Thus, the aim was to approximate the training data via a subset (coresets) or enhance training (filtering) assuming that the training and testing sets share the same distribution.

**Active learning.** There is a rich area of online active learning literature, which continually filters data while training (Ein-Dor et al., 2020; Wang et al., 2022; Yuan et al., 2020; Tamkin et al., 2022), requiring to query an annotator for more labeled data and oftentimes, rely on properties of the models in-training to select data. Here, we are interested in exploring dataset subset selection prior to training and without knowledge of model weights.

**Domain adaptation.** Domain adaptation is a parallel line of work for model specialization, where a machine learning model is trained on one domain of data (the "source" domain) and then applied to a different domain (the "target" domain), oftentimes with significant distribution shift Farahani et al. (2020). Other related work includes active domain adaptation (ADA), which combines the ideas from active learning with domain adaptation. The core idea is to actively choose a subset of unlabeled

target data points to label to most effectively adapt a model to the target domain (Huang et al., 2023; Prabhu et al., 2020). ADA focuses on actively/iteratively selecting and then collecting labels for data within the deployment during the specialization process, whereas DS3 selects data in a single-shot approach prior to specialization on an already available pool of data.

# C ADDITIONAL RESULTS

## C.1 SAMPLE EFFICIENCY

We plot the results of the baselines that had set thresholds of subset size in Figure 4 for the linear probing results. For 9 of the 16 deployments, there was no subset from the threshold baselines that outperformed using all the data. However, for the examples of AutoArborist Deployment 1, 2, and 4; GeoDE Deployment 3 and 4; and iWildCam Deployment 1 and 2, there was a simple threshold baseline that outperformed all the data. In particular, AutoArborist Deployment 1 greatly benefitted from subsetting, likely due to its extremely long-tailed nature, as shown in Figure 10. These "efficiency-style" experiments were not performed for the full-finetuning ResNet50 models, due to computational constraints of training a great number of models.

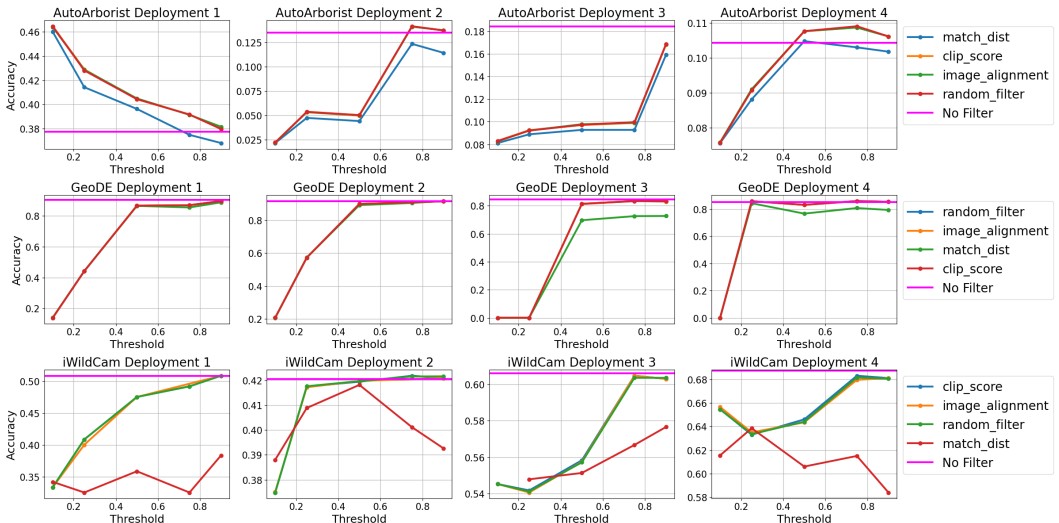

Figure 4: Plotting the sample efficiency of the baselines, for the baselines that thresholds of subset size were set (CLIP-score, Image-Align, Match-Dist), with Random as a comparison point) for the linear-probing results. NuScenes was left out because it uses MSE as a performance metric and cannot use the Match-Dist baseline because it is a regression baseline. We find that oftentimes, models perform nearly just as well with 50% of the data, with examples of certain subsets outperforming using all the data.

## C.2 OTHER MODEL EXPERIMENTS

**ViT Linear Probes:** In addition to full-finetuning of ResNets, we also performed linear probing of ViT embeddings as described in Section D. In contrast to full-finetuning, the gains made from well-chosen subsets to using all the data are minimal, with a maximum accuracy gain of $18\%$ for AutoArborist. In contrast to the full-finetuning results, there does not always exist a well-chosen subset for all deployments. We hypothesize that the model is not sufficiently updated for dataset subset selection when using linear probes - since the embeddings from the pretrained ViT model remain the same, the data does not play as much of an impact. Additionally, linear probes overall perform much worse than a ResNet full-finetune, with the notable exception of GeoDE. This is likely because GeoDE is an object recognition dataset, closer to the pretrained ImageNet-21k weights that the ViT model is well-posed to recognize. AutoArborist, iWildCam, and NuScenes are significantly OOD from the pretraining dataset of ViT and as such, linear probes perform poorly – more training/finetuning is required for optimal performance. Additionally, the scale/size of the subset is much more important

| Dataset | Metric | Deploy # | Non subset | | Knowledge-driven | Unlabeled query set | | Labeled query set | | |
|---|---|---|---|---|---|---|---|---|---|---|
| | | | Query-set | All-data | | Image-Align | Near-Nbors | CLIP-score | Match-Label | Match-Dist |
| GeoDE | Acc | Deploy 1 | 0.851 | 0.904 | 0.083 | 0.869 | 0.867 | 0.897 | **0.904** | 0.887 |
| | | Deploy 2 | 0.833 | 0.914 | 0.114 | 0.914 | 0.898 | 0.914 | 0.915 | **0.916** |
| | | Deploy 3 | 0.998 | 0.844 | 0.100 | 0.834 | 0.814 | 0.832 | **0.923** | 0.727 |
| | | Deploy 4 | 0.969 | 0.853 | 0.123 | 0.858 | 0.837 | 0.854 | **0.900** | 0.794 |
| iWildCam | Acc | Deploy 1 | 0.858 | **0.509** | 0.350 | 0.508 | 0.391 | 0.508 | 0.350 | 0.383 |
| | | Deploy 2 | 0.585 | 0.420 | 0.346 | 0.423 | 0.425 | 0.421 | **0.438** | 0.418 |
| | | Deploy 3 | 0.824 | **0.606** | 0.550 | 0.604 | 0.569 | 0.604 | 0.554 | 0.567 |
| | | Deploy 4 | 0.906 | **0.688** | 0.611 | 0.681 | 0.621 | 0.683 | 0.624 | 0.639 |
| Auto Arborist | Acc | Deploy 1 | 0.373 | 0.377 | **0.545** | 0.464 | 0.392 | 0.464 | 0.401 | 0.459 |
| | | Deploy 2 | 0.109 | 0.134 | **0.245** | 0.137 | 0.141 | 0.137 | 0.139 | 0.123 |
| | | Deploy 3 | 0.169 | 0.184 | **0.234** | 0.167 | 0.099 | 0.168 | 0.103 | 0.159 |
| | | Deploy 4 | 0.125 | 0.104 | **0.300** | 0.106 | 0.108 | 0.108 | 0.108 | 0.108 |
| NuScenes | MSE | Deploy 1 | 0.494 | 0.508 | 0.730 | **0.333** | 0.391 | 0.475 | - | - |
| | | Deploy 2 | 0.226 | 0.420 | 1.160 | **0.375** | 0.425 | 0.422 | - | - |
| | | Deploy 3 | 0.559 | 0.606 | 1.112 | **0.540** | 0.566 | 0.558 | - | - |
| | | Deploy 4 | 0.434 | 0.688 | 1.137 | 0.635 | **0.621** | 0.646 | - | - |

Table 2: Best-performing subsets across hyperparameters for baseline methods across all datasets and deployments (abbreviated as Deploy) for **ViT linear probes**. Overall accuracy is reported for the classification tasks of GeoDE, iWildCam, and Auto Arborist (greater is better) and MSE is reported for the regression task of NuScenes (smaller is better). Match-Dist and Match-Label are not applicable for NuScenes, as it is a regression task and does not have clear classes/labels for these methods. Baselines are distinguished from one another by their access to information, with each baseline having access to expert knowledge, or a labeled/unlabeled query set.

| Dataset | Metric | Deploy # | Non subset | | Knowledge-driven | Unlabeled query set | | Labeled query set | | |
|---|---|---|---|---|---|---|---|---|---|---|
| | | | Query-set | All-data | | Image-Align | Near-Nbors | CLIP-score | Match-Label | Match-Dist |
| GeoDE | Acc | Deploy 1 | 0.870 | 0.871 | 0.878 | **0.895** | 0.883 | 0.890 | 0.882 | 0.886 |
| | | Deploy 2 | 0.440 | 0.894 | 0.859 | 0.896 | 0.895 | **0.902** | 0.900 | 0.882 |
| | | Deploy 3 | 0.961 | 0.758 | **0.994** | 0.837 | 0.799 | 0.894 | 0.83 | 0.879 |
| | | Deploy 4 | 0.821 | 0.858 | **0.958** | 0.779 | 0.833 | 0.882 | 0.841 | 0.830 |
| iWildCam | Acc | Deploy 1 | 0.703 | 0.635 | 0.641 | 0.562 | 0.503 | 0.505 | 0.736 | **0.753** |
| | | Deploy 2 | 0.730 | 0.332 | 0.344 | 0.433 | 0.460 | 0.460 | 0.350 | **0.494** |
| | | Deploy 3 | 0.434 | 0.723 | 0.743 | 0.537 | 0.456 | 0.421 | 0.733 | **0.755** |
| | | Deploy 4 | 0.444 | 0.650 | 0.689 | 0.600 | 0.601 | 0.156 | 0.567 | **0.732** |
| AutoArborist | Acc | Deploy 1 | 0.154 | 0.450 | **0.888** | 0.381 | 0.390 | 0.388 | 0.665 | 0.777 |
| | | Deploy 2 | 0.233 | 0.489 | **0.945** | 0.133 | 0.141 | 0.138 | 0.645 | 0.564 |
| | | Deploy 3 | 0.138 | 0.169 | **0.384** | 0.179 | 0.129 | 0.169 | 0.156 | 0.234 |
| | | Deploy 4 | 0.147 | 0.155 | **0.452** | 0.104 | 0.118 | 0.146 | 0.152 | 0.233 |
| NuScenes | MSE | Deploy 1 | 0.061 | 0.034 | **0.034** | 0.044 | 0.055 | 0.072 | - | - |
| | | Deploy 2 | 0.071 | **0.011** | 0.045 | 0.167 | 0.056 | 0.033 | - | - |
| | | Deploy 3 | 0.099 | 0.045 | **0.034** | 0.041 | 0.122 | 0.072 | - | - |
| | | Deploy 4 | 0.120 | 0.049 | **0.040** | 0.084 | 0.399 | 0.050 | - | - |

Table 3: Best-performing subsets across hyperparameters for baseline methods across all datasets and deployments (abbreviated as Deploy) for **ViT LoRA finetuning**. Overall accuracy is reported for the classification tasks of GeoDE, iWildCam, and Auto Arborist (greater is better) and MSE is reported for the regression task of NuScenes (smaller is better). Match-Dist and Match-Label are not applicable for NuScenes, as it is a regression task and does not have clear classes/labels for these methods. Baselines are distinguished from one another by their access to information, with each baseline having access to expert knowledge, or a labeled/unlabeled query set.

with linear probing than with full-finetuning – this is perhaps why the query sets and expert subsets (which are quite small relative to the entire training dataset) perform poorly relative to training on all data. In contrast, a ResNet50 does not need to be trained on much data (such as the query/expert subsets) to perform near-optimal, if the data is nearly in-distribution.

**ViT LoRA Finetuning:** We also performed ViT LoRA finetuning as described in Section D. We see very similar results to the ResNet full-finetuning, largely with greater performance on some deployments than with ResNet (likely due to the larger model architecture).

## D ADDITIONAL TRAINING INFORMATION

**(1) Subset selection step:** Users must select data from the general training pool for a deployment with the given query set. Query sets are split from the evaluation/deployment set with a stratified sampling strategy, ensuring that at least two of each class is in the query set. This is done to ensure that long-tailed datasets (such as iWildCam and AutoArborist) have all classes present in the query set to be representative of the deployment. iWildCam, NuScenes, and GeoDE have 500 sample query

sets, whereas AutoArborist has a 1500 sample query set (because of its extremely long-tailed nature and the complexity of the dataset).

**(2) Training/Fine-tuning step:** For all classification datasets, we use ResNet50 (He et al., 2015) for full-finetuning, a ViT (Dosovitskiy et al., 2020) for LoRA finetuning and for linear probes of the training subsets, chosen for efficiency due to the number of baselines. No additional data augmentation was performed to fully understand the role of the subset quality by itself. For full-finetuning, we use the Adam optimizer with cross-entropy loss, starting from ImageNet-1k weights and updating all layers. For the linear probing, we train a `sk-learn` logistic/linear regression classifier on top of ViT-B/32 features pretrained from ImageNet-21k and labels for classification/regression tasks, respectively. For LoRA finetuning, we use the `peft` package to select layers to finetune/ We run a small hyperparameter sweep for each baseline across batch sizes $\{32, 64, 128\}$ and learning rates $\{0.01, 0.001, 0.0001\}$ for each deployment across a validation set, split from the training subset in a random $90/10$ split. All models were trained on A100 GPU's. All models were implemented using PyTorch.

For detection tasks, we use YOLOv8n (Jocher et al., 2023) model with default training parameters, such as learning rates/schedules and default COCO-pretrained models, for 100 epochs, and subsample images to 640p resolutions, chosen for efficiency due to computational costs of training detectors. These models were trained across 4 L40 GPU's in parallel.

# E  ADDITIONAL DATASET DETAILS

## E.1  IWILDCAM

**Additional dataset information.** These images tend to be taken in short bursts following the motion-activation of a camera trap, so the images can be additionally grouped into sequences of images from the same burst, though our baseline models do not exploit this information, and our evaluation metric treats each image individually. However, a grouped sequence is in the same split of the data (train, test, query) in order to avoid model memorization. Each image is associated with the following metadata: camera trap ID, sequence ID, and datetime.

**Deployment splits.**    The splits for each deployment were created by mapping the locations of the iWildCam camera traps to latitude and longitude, then clustering geospatially, where every cluster formed a 100km radius. The deployment splits are subsets of the testing split of the WILDS-iWildCam (Koh et al., 2020) dataset. In  (Koh et al., 2020), the "in-distribution" test split (`test-id`) was split from the training set by the time of the capture, whereas the "out-of-distribution" test split (`test-ood`) where was split from the training set by location. Deployments 1 and 3 are from the original `test-id` split of the dataset and Deployments 2 and 4 are from the `test-ood` split. As a result, Deployments 1 and 3 are ID dataset subselection tasks – meaning that data from the locations in these deployments exists in the training pool. In contrast, Deployments 2 and 4 are OOD dataset subselection tasks – there is no existing data from these locations in the training pool. We expect that OOD tasks will be more difficult to find well-performing subsets for. Additional details about each deployment are given in Table 4 and visualized in Figure 6 and Figure 5.

**Expert subsets.**    For Deployments 1 and 3 (the ID deployments), the expert subsets were created by only choosing data from the relevant locations in the testing pool (these locations are given in Table 4) and eliminating irrelevant classes that are not present in the testing set. For Deployments 2 and 4 (the OOD deployments), the expert subsets were created by only choosing data from locations that are within 500km of the locations in the deployments and eliminating irrelevant classes that are not present in the testing set.

| Deployment # | # classes | # images | locations | class_label |
|---|---|---|---|---|
| Deployment 1 | 14 | 1744 | 225, 333, 312, 530, 416, 541, 25, 356, 504, 521, 71, 202, 516, 224, 162 | funisciurus carruthersi, cricetomys gambianus, cephalophus nigrifrons, protoxerus stangeri, atherurus africanus, turtur tympanistria, francolinus nobilis, potamochoerus larvatus, cercopithecus lhoesti, pan troglodytes, cercopithecus mitis, francolinus africanus, hylomyscus stella, canis lupus |
| Deployment 2 | 15 | 4348 | 167, 282, 385, 123, 110, 144, 231, 417, 314 | francolinus nobilis, cephalophus nigrifrons, empty, funisciurus carruthersi, atherurus africanus, cricetomys gambianus, cercopithecus lhoesti, genetta tigrina, cercopithecus mitis, paraxerus boehmi, pan troglodytes, genetta servalina, mus minutoides, cephalophus silvicultor, turtur tympanistria |
| Deployment 3 | 30 | 2558 | 242, 51, 101, 508, 372, 522, 247, 138, 359, 379, 91, 157, 410, 259, 369, 73, 220, 415, 518, 387, 10, 170, 226, 443, 383, 251, 450, 532, 503, 81, 437, 151, 127, 113, 471, 197, 112, 130, 65, 273, 254, 366, 400, 31, 551, 298, 24, 124, 169, 318, 159 | empty, aepyceros melampus, madoqua guentheri, ichneumia albicauda, kobus ellipsiprymnus, syncerus caffer, leptailurus serval, bos taurus, giraffa camelopardalis, loxodonta africana, equus grevyi, crocuta crocuta, equus quagga, capra aegagrus, felis silvestris, panthera pardus, camelus dromedarius, proteles cristata, phacochoerus africanus, acryllium vulturinum, papio anubis, ovis aries, hippopotamus amphibius, tragelaphus scriptus, unknown bird, eupodotis senegalensis, tragelaphus strepsiceros, lepus saxatilis, tragelaphus oryx, lissotis melanogaster |
| Deployment 4 | 33 | 13931 | 501, 306, 206, 499, 131, 53, 140, 375, 179, 193, 291, 229, 249, 108, 477, 328, 152, 302, 278, 329, 422, 540, 520, 136, 489, 492 | bos taurus, loxodonta africana, equus quagga, aepyceros melampus, empty, phacochoerus africanus, oryx beisa, tragelaphus oryx, lycaon pictus, syncerus caffer, crocuta crocuta, madoqua guentheri, acryllium vulturinum, capra aegagrus, caracal caracal, lepus saxatilis, hippopotamus amphibius, giraffa camelopardalis, papio anubis, camelus dromedarius, hystrix cristata, equus grevyi, panthera leo, ovis aries, panthera pardus, proteles cristata, hyaena hyaena, canis lupus, canis mesomelas, leptailurus serval, kobus ellipsiprymnus, eupodotis senegalensis, lissotis melanogaster |

Table 4: iWildCam Deployment split details. As seen, the deployments have severe label shift from one another and come from disjoint locations. They also vary in size and the number of classes.

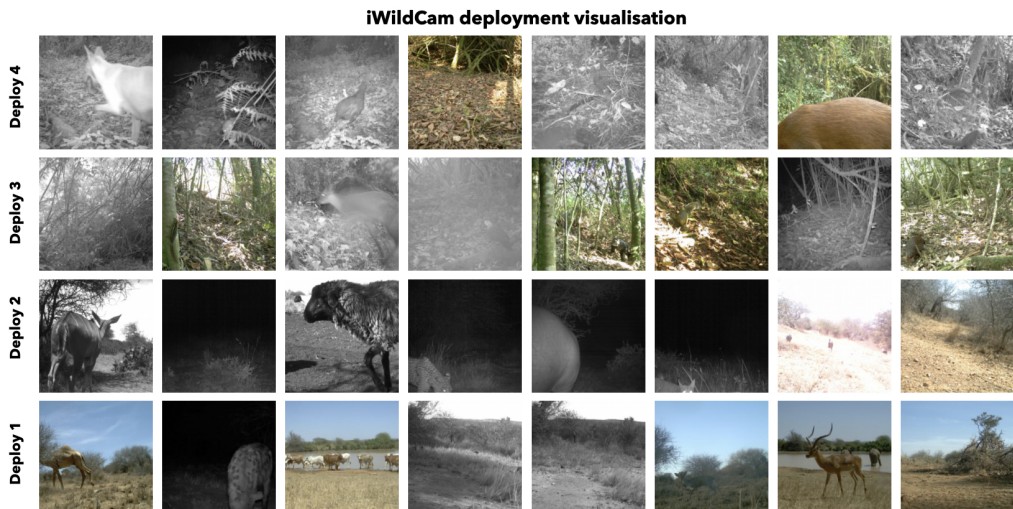

Figure 5: Visualization of the iWildCam dataset across deployments

## E.2   GEODE

**Additional dataset information.** Datasets used for object classification tasks are often constructed by scraping images from the web. Examples of such datasets include ImageNet (Deng et al., 2009a; Russakovsky et al., 2014), Open Images (Kuznetsova et al., 2018), PASS (Asano et al., 2020), CLIP-400M (Brown et al., 2020) etc. Constructing such datasets is cheap, and thus scalable, however, such datasets are known to contain, among others, geographical biases (Shankar et al., 2017). Rather than scraping images from the web, GeoDE (Ramaswamy et al., 2023) crowdsources a dataset roughly balanced across 40 different objects and 6 world regions. Crowdsourcing a dataset allows for tighter control over the data distribution. For example, it allows us to target specific regions and objects that are underrepresented within webscraped datasets. However, it can also be prohibitively expensive, limiting the size of such datasets. Thus, it becomes paramount to understand which objects and regions should be targeted within crowdsourced data collection.

**Deployment splits.** There are two types of deployments: country and object splits. As mentioned, Deployments 1 and 2 are Nigeria and Indonesia, respectively, chosen because they are the two countries with the regions with the poorest performance in the original GeoDE paper (Ramaswamy et al., 2023). Similarly, Deployments 3 and 4 are of indoor/outdoor objects, respectively, with the worst performing classes chosen for this deployment. In the benchmark, we also include an additional potential pool of training data not explored in this experimental study – the "low data quality" pool not included in the original GeoDE paper. Users of this benchmark are allowed to add this data to their subset. The deployment's images are visualized in Figure 7.

**Expert subsets.** For Deployments 1 and 3 (the ID deployments), the expert subsets were created by only choosing data from the relevant locations in the testing pool (these locations are given in Table 4) and eliminating irrelevant classes that are not present in the testing set. For Deployments 2 and 4 (the OOD deployments), the expert subsets were created by only choosing data from locations that are within 500km of the locations in the deployments and eliminating irrelevant classes that are not present in the testing set.

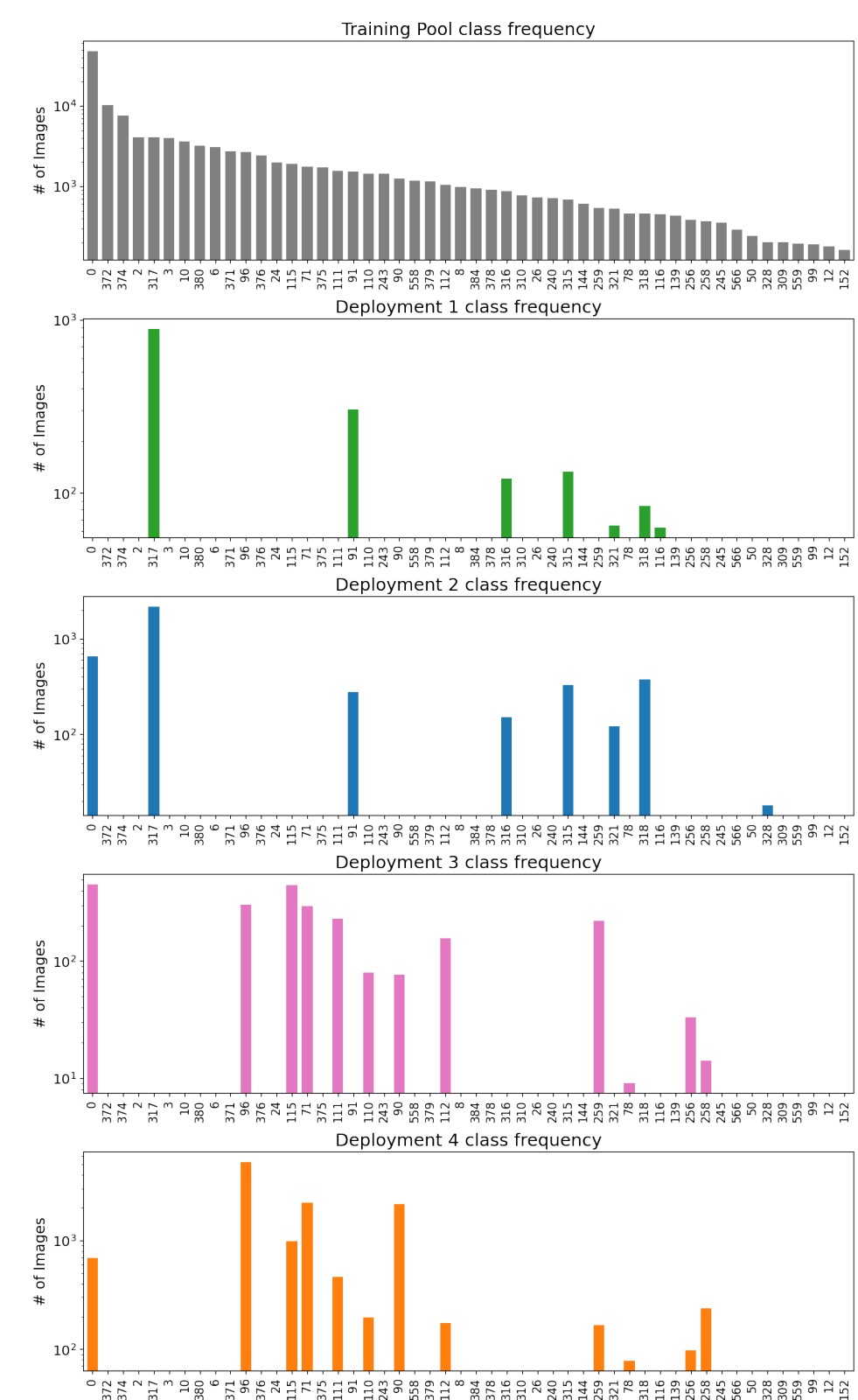

Figure 6: iWildCam deployment label distribution for the 50 most common classes (common determined by the training pool) in log-scale. As seen, there is significant label shift from the training pool to the deployments, and from the deployments to each other.

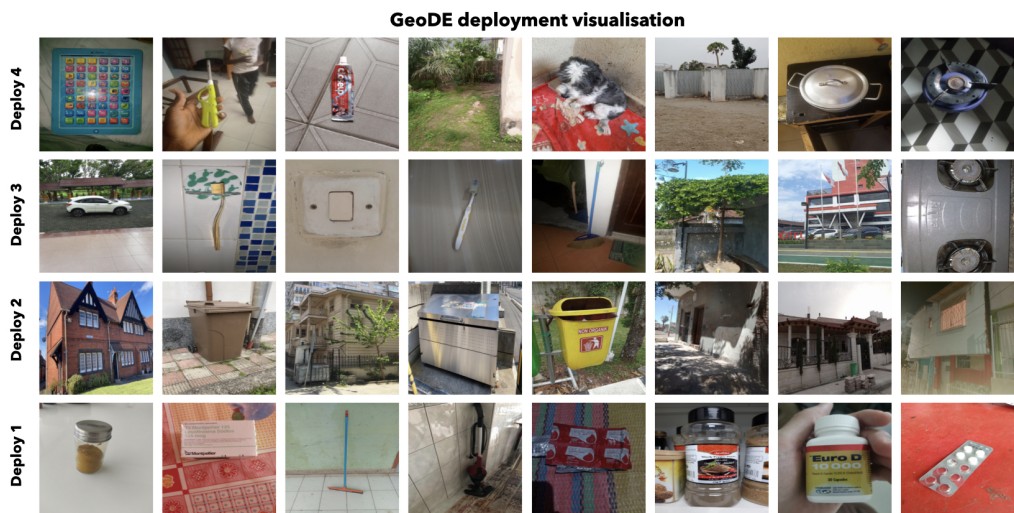

Figure 7: Visualization of the GeoDE dataset across deployments

| Deployment # | # classes | # images | class_label |
|---|---|---|---|
| (1) Nigeria | 40 | 2155 | bag, hand soap, dustbin, toothbrush, toothpaste toothpowder, hairbrush comb, chair, hat, light fixture, light switch, plate of food, spices, stove, cooking pot, cleaning equipment, lighter, medicine, candle, toy, jug, streetlight lantern, front door, tree, house, backyard, truck, waste container, car, fence, road sign, dog, wheelbarrow, religious building, stall, boat, monument, flag, bus, storefront, bicycle |
| (2) Indonesia | 40 | 4348 | bag, hand soap, dustbin, toothbrush, toothpaste toothpowder, hairbrush comb, chair, hat, light fixture, light switch, plate of food, spices, stove, cooking pot, cleaning equipment, lighter, medicine, candle, toy, jug, streetlight lantern, front door, tree, house, backyard, truck, waste container, car, fence, road sign, dog, wheelbarrow, religious building, stall, boat, monument, flag, bus, storefront, bicycle |
| (3) Indoor Objects | 2 | 924 | house, waste container |
| (4) Outdoor Objects | 3 | 2116 | spices, cleaning equipment, medicine |

Table 5: GeoDE Deployment split details. As seen, the deployments have label shift from one another, varying in size and class labels.

### E.3 AUTOARBORIST

**Additional dataset information.** Environmental monitoring and Earth observation from aerial imagery have the potential to enable policymakers to make data-informed decisions to facilitate societal adaptation to a changing climate (Brandt et al., 2016). However, aerial/street-level data repositories from satellite and low-flying aircraft are currently in the petabyte scale and growing, making extracting useful and relevant information to support policy intractable without automation. Tree image classification has potential impact on humanitarian aid and disaster relief, wilderness forests, agriculture, and urban mapping with uses in city planning, resource management, and environmental monitoring. For example, urban ecologists need to know the location and type of trees in cities so that they can target replanting to improve climate adaptation. Collecting this information from ground-level tree censuses is both time-consuming and expensive, thus automated tree genus classification from Global Positioning System (GPS)-registered aerial imagery is increasingly of

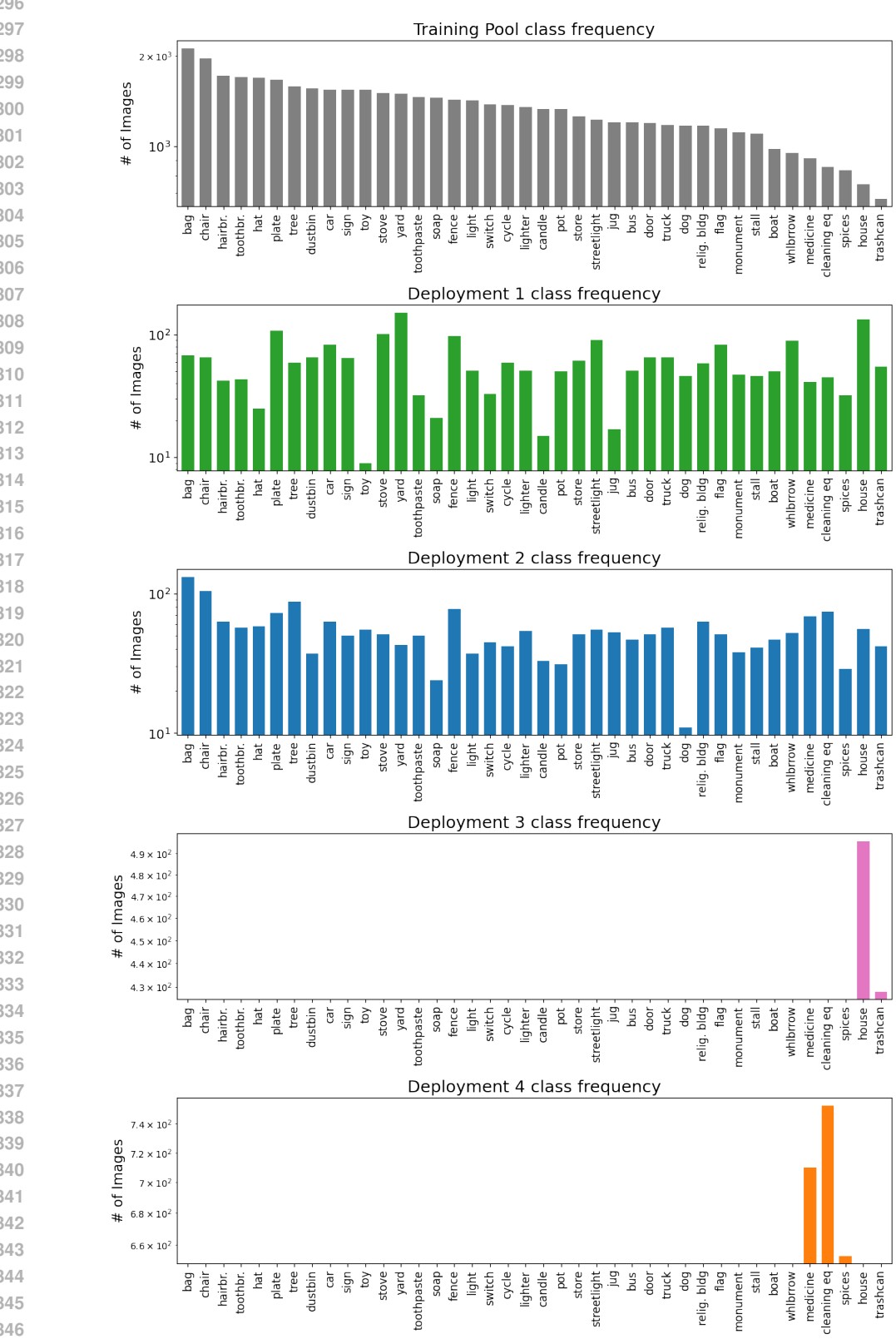

Figure 8: GeoDE deployment label distribution for all classes. As seen, there is significant label shift from the training pool to the deployments, and from the deployments to each other.

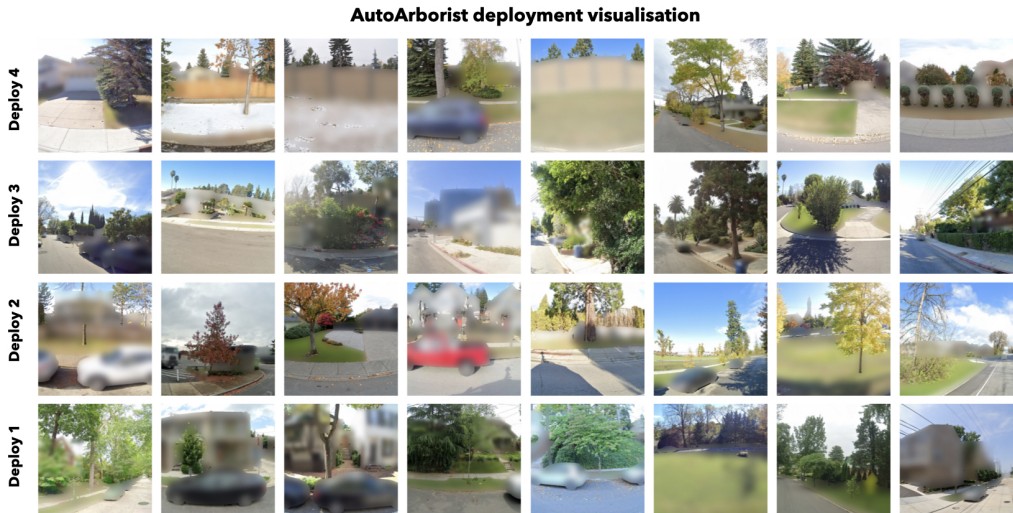

Figure 9: Visualization of the AutoArborist dataset across deployments

interest. Datasets like AutoArborist enable the computer vision community to investigate automated methods for tree genus classification from aerial imagery at scale, containing images and genus labels for over 1M individual trees (Beery et al., 2022b).

Additional challenges in the data include:

- **Noisy labels.** Images are commonly mislabeled: with genus classification is difficult and requires specialized expertise, GPS localization from the ground can be in error, there are often multiple trees within a single image with only a single label, and temporal inconsistencies can occur as trees are not imaged and labeled at the same time.

- **Non-IID data.** Geospatial data also breaks the typical deep learning assumption that data will be independent and identically distributed (IID) spatially close examples often contain correlations. For example, trees are often planted in groups (e.g. a row of cherry trees along the same street).

- **Fine-grained and long-tailed class distribution.** Tree classification is fine-grained, with only subtle differences between many genera, and the distribution of trees is long-tailed. These characteristics tend to skew classification models towards predicting predominant classes.

- **Geospatial distribution shift.** Finally, this dataset contains significant covariate and subpopulation distribution shift due to variations in weather, differences in urban planning specific to each city, and temporal changes at different locations.

**Deployment splits.** The splits for each deployment were created by city, where the evaluation splits are subsets of the testing split of the original AutoArborist (Beery et al., 2022b) dataset. Deployments 3 and 4 (Los Angeles and Washington DC) are the "in-distribution" (ID) deployments, where the training data from these cities was left in the training pool. Deployments 1 and 2 (Surrey and Calgary) were the "out-of-distribution" (OOD) deployments, where there was no data from these cities in the training pool. The training pool additionally contains data from 19 other major cities in North America.

**Expert subsets.** For expert subsets, we used data from the city itself for the ID deployments, and data from the closest cities for the OOD deployments. Accordingly, we used data from San Francisco and San Jose for Los Angeles and Charlottesville, Pittsburgh, and New York for Washington DC. We expect that OOD tasks will be more difficult to find well-performing subsets for in the benchmark. Label distribution shift is visualized in Figure 10 and covariate shift visualized in Figure 9.

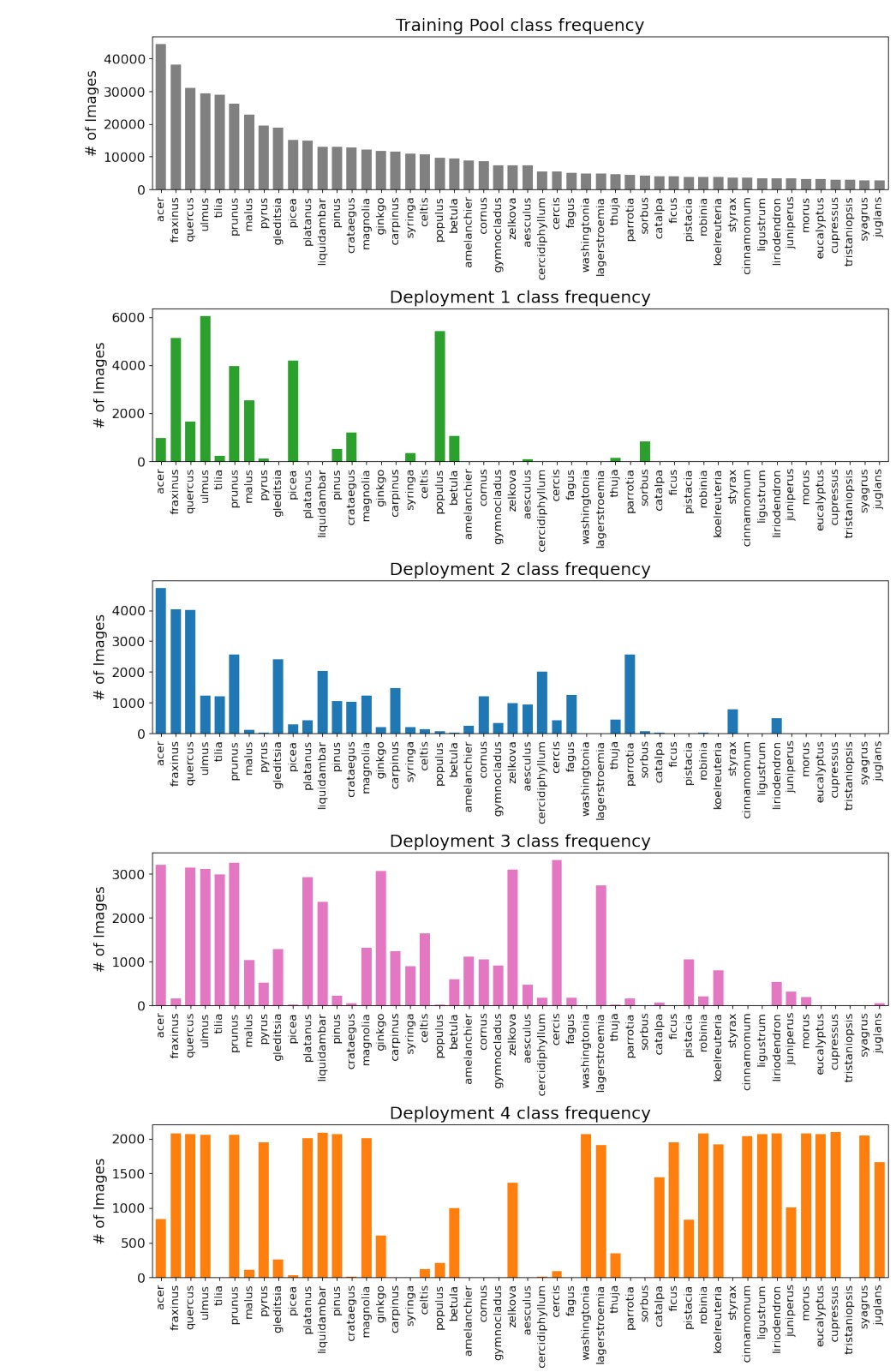

Figure 10: AutoArborist deployment label distribution for the 50 most common classes (commonly determined by the training pool) in log-scale. As seen, there is a significant label shift from the training pool to the deployments, and from the deployments to each other.

| Deployment | # classes | # imgs | class_label |
|---|---|---|---|
| (1) Surrey | 66 | 44295 | cladrastis, pinus, styrax, cornus, prunus, quercus, crataegus, liquidambar, ulmus, gleditsia, fraxinus, tilia, parrotia, platanus, stewartia, acer, aesculus, fagus, metasequoia, carpinus, zelkova, celtis, magnolia, liriodendron, amelanchier, cercidiphyllum, davidia, nyssa, syringa, pseudotsuga, thuja, cercis, phellodendron, chamaecyparis, ginkgo, calocedrus, gymnocladus, sorbus, sequoiadendron, picea, robinia, taxodium, catalpa, abies, malus, tsuga, eucommia, alnus, koelreuteria, betula, pyrus, populus, cedrus, sequoia, salix, juglans, pistacia, hibiscus, halesia, corylus, nothofagus, ilex, thujopsis, pterocarya, larix, paulownia |
| (2) Calgary | 15 | 36347 | acer, ulmus, syringa, crataegus, sorbus, fraxinus, picea, prunus, populus, quercus, betula, malus, thuja, larix, tilia, pinus, elaeagnus, salix, pyrus, pseudotsuga, aesculus, abies, hippophae, caragana, juniperus, ribes, cotoneaster, alnus, tsuga, gleditsia |
| (3) Los Angeles | 30 | 146849 | platanus, cedrus, ailanthus, nerium, syagrus, juglans, pinus, cupaniopsis, washingtonia, schinus, cinnamomum, lagerstroemia, jacaranda, libocedrus, syzygium, ceratonia, podocarpus, morus, liquidambar, zelkova, yucca, melaleuca, acacia, laurus, pistacia, fraxinus, robinia, magnolia, thuja, tabebuia, ligustrum, catalpa, koelreuteria, ulmus, salix, pittosporum, quercus, trachycarpus, casuarina, betula, rhus, callistemon, calocedrus, prunus, olea, archontophoenix, tristania, cassia, eucalyptus, citrus, lagunaria, ficus, phoenix, liriodendron, cordyline, malus, pyrus, celtis, cupressus, brachychiton, alnus, acer, ginkgo, juniperus, hymenosporum, photinia, eriobotrya, hibiscus, bauhinia, melia, thevetia, geijera, sequoia, sequoiadendron, maytenus, grevillea, erythrina, broussonetia, carya, tipuana, cercis, chionanthus, calodendrum, ceiba, chamaerops, sapium, diospyros, albizia, gleditsia, sambucus, musa, araucaria, strelitzia, vitex, psidium, cocos, dodonaea, metrosideros, heteromeles, populus, macadamia, sphaeropteris, eugenia, leptospermum, feijoa, platycladus, persea, casimiroa, sophora, dracaena, xylosma, livistona, schefflera, crinodendron, brahea, leucaena, ilex, arbutus, taxodium, punica, tamarix, butia, agonis, harpephyllum, nicotiana, rhaphiolepis, crataegus, plumeria, cycas, cornus, euonymus, lycianthes, myoporum, parkinsonia, picea, cercidiphyllum, elaeagnus, euphorbia, viburnum, quillaja, cotoneaster, pyracantha, paulownia, cocculus, caesalpinia, camellia, stenocarpus, lyonothamnus, maclura, osmanthus, beaucarnea, firmiana, castanea, umbellularia, wisteria, sorbus, metasequoia, myrtus, ziziphus, hakea, spathodea, annona, cryptomeria, olmediella, solanum, abies, aesculus, howea, ensete, carica, pseudotsuga, fremontodendron, chiranthodendron, chamaecyparis, cotinus |
| (4) Washington DC | 33 | 71519 | acer, tilia, syringa, celtis, gleditsia, ginkgo, cercis, amelanchier, platanus, malus, zelkova, prunus, nyssa, liquidambar, magnolia, lagerstroemia, pinus, ulmus, gymnocladus, quercus, carpinus, cladrastis, pyrus, betula, robinia, juniperus, koelreuteria, ilex, metasequoia, liriodendron, taxodium, cornus, styphnolobium, pistacia, morus, fagus, cedrus, aesculus, populus, crataegus, chionanthus, fraxinus, parrotia, laburnum, cercidiphyllum, rhus, catalpa, eucommia, halesia, ostrya, stewartia, sassafras, picea, cryptomeria, thuja, juglans, diospyros, cotinus, ailanthus, carya, oxydendrum, tsuga, salix, maclura, phellodendron, maackia, paulowniar |

Table 6: AutoArborist Deployment split details. As seen, the deployments have severe label shift from one another and come from disjoint locations. They also vary in size and the number of classes.

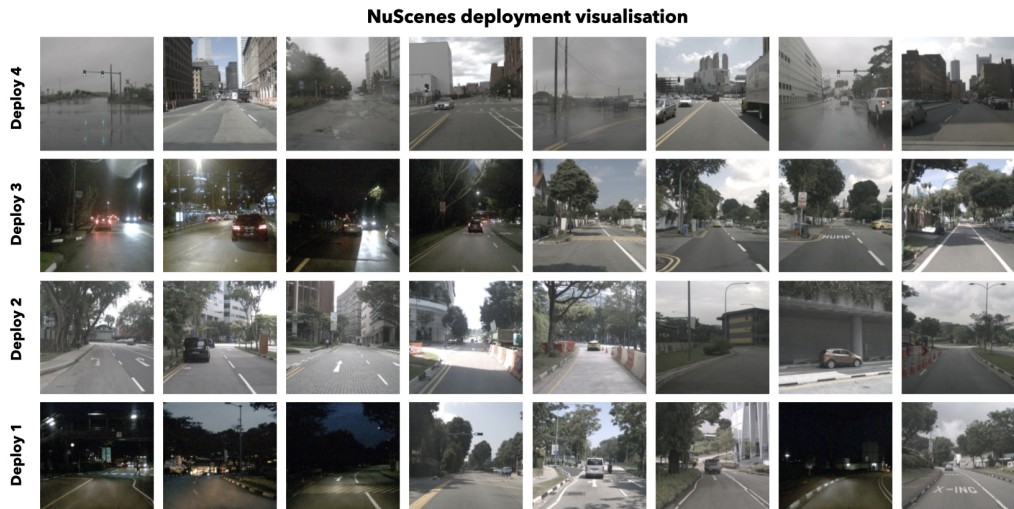

Figure 11: Visualization of the NuScenes dataset across deployments

### E.4 NUSCENES

**Additional data information.** The images were captured from a video stream recorded while driving a car. Each image is paired with a steering angle control from the CAN bus, synchronized with the sensor timestamps of both the camera and CAN bus data. To label each image with the correct steering angle, we apply 1D interpolation to create a continuous function of the steering angle and query it based on the camera's timestamp. The steering angle, measured in radians, ranges from -7.7 to 6.3, with 0 indicating straight driving, positive values indicating left turns, and negative values indicating right turns. To ensure alignment between images and steering control data, samples with vehicle velocities below 1 m/s are removed.

**Deployment splits.** The splits for each deployment were created by city, where the evaluation splits are subsets of the testing split of the original NuScenes (Caesar et al., 2020) dataset. All deployments are in-distribution, meaning that there exists data from each city's deployment in the training pool. In contrast to AutoArborist, iWildCam, and GeoDE, where there is "extraneous" data in the training pool (that isn't necessarily relevant to the deployment geospatially or label-wise), there only exists data from the deployment cities in the training pool. The covariate shift in this dataset is visualized in Figure 11.

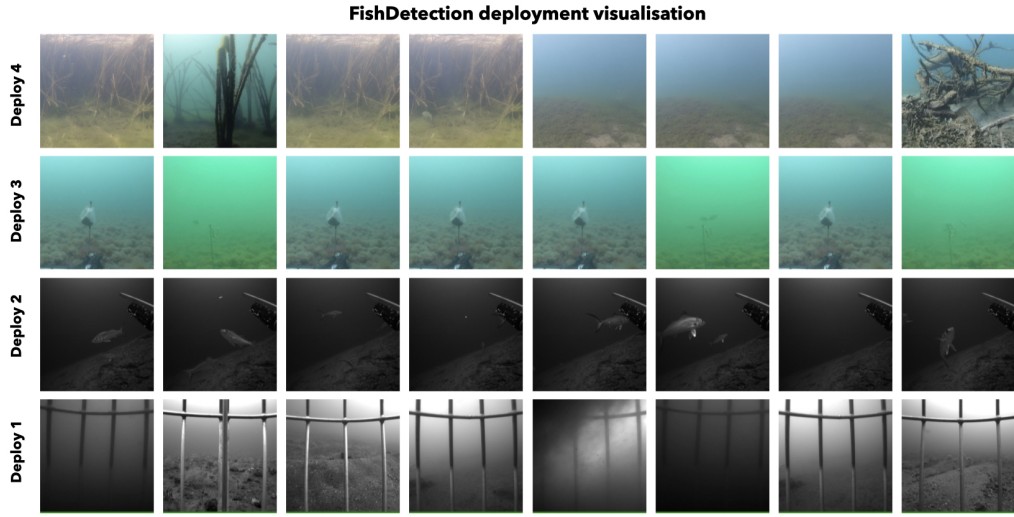

Figure 12: Visualization of the FishDetection dataset across deployments