# OpenReview forum: "DataS3: Dataset Subset Selection for Specialization"
_ICLR.cc/2026/Conference — Submitted to ICLR 2026_

### Official Review · Reviewer_mjnB · 2025-10-28

**Soundness:** 3
**Presentation:** 3
**Contribution:** 3
**Rating:** 4
**Confidence:** 3

**Summary:**

This paper introduces the problem of dataset subset selection for specialization (DS3), where the goal is to select a subset of a training dataset that optimizes deployment performance. The authors claim that specific deployments often have unique data distributions, which often diverge from general training set distributions. Thus, this paper introduces a benchmark to evaluate the performance of existing subset selection methods on real-world deployment tasks.

**Strengths:**

1. The benchmark demonstrates the novelty. This paper first investigates a real-world application problem, dataset subset selection for specialization (DS3), which aims to select training data tailored to improve performance on real-world deployment tasks. Due to the complexity of the deployment environment, the authors propose using a query set drawn from the target deployment-specific distribution to effectively guide the selection of training data.

2. The benchmark is comprehensive. This benchmark covers five datasets with various tasks, including classification, detection, and regression. In addition, the authors construct diverse applications of ML for each dataset and provide a proof-of-concept subset, which demonstrates the usefulness of subset selection. The authors also compare the performance of dataset subset selection algorithms across the benchmark, across different scenarios.

**Weaknesses:**

1. How to define the deployment-specific distribution. The authors claim that models need to perform well on specific deployments rather than the domain. As illustrated in Figure 3, the distinctions between different deployment applications lie in their spatial environments.  It is clear that different real-world scenarios exhibit distribution shifts. For developers, it is often straightforward to select training samples from the target deployment environment. For example, if the task is to perform species classification in Southern Africa, one can select training data collected from that region. The authors are encouraged to further elaborate on the unique challenges and characteristics arising from real-world application deployment.

2. The challenge presented by the benchmark remains unclear. As shown in Table 1, knowledge-driven subsets generally achieve the best performance across different datasets and deployment settings. These subsets are typically constructed by selecting training data from regions relevant to the target locations. The findings suggest that the main challenge of the benchmark lies in selecting training data from scenarios relevant to the deployment environment. This can be easily achieved by selecting training data based on the metadata of datasets.

3. Lack of new methods. While the benchmark is useful, the paper primarily evaluates existing selection strategies on the benchmark. The authors are encouraged to propose novel algorithms tailored to DS3.

**Questions:**

1. In real-world scenarios, how can query sets be constructed? One possible approach is to collect training data directly from the deployment environment if query sets can be constructed.

2. How do query sets affect the performance of the selection method? If the query sets are biased, does the performance of the selection algorithm on the model depend on the quality of the query sets?

---

### Official Review · Reviewer_QPN6 · 2025-11-01

**Soundness:** 3
**Presentation:** 2
**Contribution:** 1
**Rating:** 2
**Confidence:** 4

**Summary:**

The paper introduces the problem of Dataset Subset Selection for Specialization (DS3), which aims to select subsets of training data that best support performance on specific deployment distributions, rather than general domains. The authors present DATAS³, the first benchmark and dataset suite specifically designed for this problem, spanning five real-world application domains with distinct deployment scenarios. Through extensive experiments, the paper shows that models trained on general-purpose data often underperform on deployment-specific tasks, while expert-curated subsets can yield improvements. The work provides a good empirical foundation for studying deployment-aware data selection and emphasizes the growing importance of targeted data curation as machine learning models move from broad pretraining to specialized real-world deployments.

**Strengths:**

- The paper makes an effort to establish what appears to be the first benchmark framework for evaluating dataset subset selection methods tailored to real-world deployment scenarios.

- It presents use cases across multiple domains and benchmark datasets, and the authors plan to release these resources, which could benefit the community.

- The proposed method demonstrates consistent performance improvements across the selected baselines and datasets within the benchmark.

**Weaknesses:**

The overall methodological or mathematical novelty appears somewhat limited for a top-tier venue like ICLR. However, the work could make a valuable contribution as part of a benchmark or empirical study track, given its focus on systematic evaluation and dataset development.

- There is substantial prior work on targeted subset selection addressing similar objectives, but these papers are not adequately discussed.
e.g. *Submodular Mutual Information for Targeted Data Subset Selection* (Suraj Kothawade et al.), *LESS: Selecting Influential Data for Targeted Instruction Tuning* (Mengzhou et al.), *PRISM: A Rich Class of Parameterized Submodular Information Measures for Guided Subset Selection*.

The Related Work section thus seems limited. I saw Appendix, a little bit detail into active learning and other targetted selection techniques in the main paper will benefit.

- The baseline choice is relatively weak — it is unclear why active learning or other stronger baselines were not included for comparison.

- The proposed method lacks clarity in its formulation: the algorithmic details of the selection process are not well explained.

- It is unclear whether the loss function in Eq. (1) is convex and what the exact search space and computational complexity are. These aspects should be formally specified for reproducibility and completeness.

**Questions:**

- Could the authors clarify what exactly is meant by the expert-driven subset? Is it being treated as a form of ground truth or reference subset for comparison?
- It would be helpful to include an analysis of how performance varies with pruning size or subset size, and how this trend compares across different baselines on the same dataset.
- See last point in weakness.

---

### Official Review · Reviewer_URgX · 2025-11-01

**Soundness:** 4
**Presentation:** 4
**Contribution:** 3
**Rating:** 6
**Confidence:** 4

**Summary:**

Foundational models require a large amount of data and especially when deployed in specific scenarios often struggle to make proper prediction. Deployment have unique characteristics i.e distribution imbalance. This requires us to curate specialised data for deployment and this is where the power of DS3 comes.

**Strengths:**

Novel Problem Formulation: The paper addresses an important practical challenge - how foundational models struggle with deployment-specific characteristics and distribution imbalances. The DS3 framework for specialization is a valuable contribution to the community.

Practical Relevance: The focus on dataset curation for specialized deployment scenarios (hospitals, national parks, etc.) addresses a real-world need where generic foundational models underperform.

Methodological Contribution: The introduction of the DS3 dataset and framework provides a useful benchmark for the research community.

**Weaknesses:**

Lack of Clarity on Specialization Scenarios: The introduction could benefit from more concrete examples of the "unique characteristics" mentioned. What specific challenges arise in hospitals vs. national parks? More detailed characterization would strengthen the motivation.

(Probable ? ) Discussion of Related Work: The paper doesn't adequately address why existing domain generalization techniques (mixup, selective augmentation) [1,2] wouldn't be applicable to DS3 selection. Given that works like "Improving Out-of-Distribution Robustness via Selective Augmentation" and "C-Mixup" show promise for domain transfer, this omission seems significant.
[1]Improving Out-of-Distribution Robustness via Selective Augmentation
[2]C-Mixup: Improving Generalization in Regression,


Unclear Relationship to Existing Methods: Contribution (iii) appears to overlap significantly with established coreset/subset selection problems. The novelty over existing approaches needs clearer articulation.

Notation Error: Line 117 in the unlabeled section appears to have incorrect notation - should this be y_1, ..., y_m instead?

Evaluation Metrics: The standard loss formulation comparison is confusing. If accuracy on the test set is lower than training on the whole dataset, how do we assess whether DS3 is actually improving performance for specialized deployment?

Unlabeled Query Set: The value proposition of the unlabeled aspect isn't clear. Why not use targeted deployment with random subset selection combined with established domain adaptation techniques?

Limited Model Scope: Are the \theta parameters restricted to ResNet/ViT models? Have the authors considered using these as feature extractors with simpler methods like k-means clustering or other coreset techniques for comparison ?

**Questions:**

1)Can you provide more concrete examples of deployment specialization scenarios?
2)How does DS3 compare to combining random subset selection with domain adaptation techniques?
3)What makes this approach superior to existing coreset methods?
4)Could you clarify the evaluation framework when subset selection intentionally reduces overall accuracy?

5)More baseline comparisons with established subset selection methods would strengthen the experimental validation ? I am not necessarily saying to add more experiments but comparing to coreset/subset selection methods would be really useful.

---

### Official Review · Reviewer_BMz4 · 2025-11-01

**Soundness:** 2
**Presentation:** 3
**Contribution:** 2
**Rating:** 2
**Confidence:** 4

**Summary:**

This study focuses on the data selection problem in domain adaptation (DA) and constructs a benchmark dataset that enables systematic evaluation of this challenge. Specifically, the authors developed a benchmark that incorporates expert knowledge, covering five datasets and twelve distinct deployment environments. Using this benchmark, they conducted comparative experiments under three conditions: (1) when data selection is performed by human experts, (2) when only unlabeled data from the deployment environment are available, and (3) when a small amount of labeled data from the deployment environment is available. The results demonstrate that expert selection achieves the highest performance, while existing automatic data selection methods perform poorly under tasks (2) and (3). These findings quantitatively reveal the limitations of current approaches. This work clarifies the challenges of data selection in real-world deployment scenarios and provides a valuable foundation for future research on domain adaptation.

**Strengths:**

This paper addresses the problem of data subset selection in domain adaptation, an issue that has become increasingly important in the era of foundation models. By focusing on data selection challenges under realistic deployment scenarios, it provides practical value through its clear articulation of the problem's significance.

The construction of a benchmark dataset that incorporates expert knowledge and covers five datasets and twelve deployment environments is particularly noteworthy. This benchmark can serve as an objective evaluation foundation for future research in domain adaptation and represents a valuable contribution to the broader machine learning community.

The paper is clearly written, with its research objectives, methodology, experimental design, and interpretation of results presented in a well-organized and logical manner. Its structure enables readers to easily grasp the significance and contributions of the study.

**Weaknesses:**

Although Domain Adaptation (DA) has been an active research area for many years, the related work section of this paper does not appear to sufficiently cover the major research trends and existing benchmarks in the field. It would be desirable to more clearly position the newly constructed benchmark by situating it in relation to existing DA datasets and problem settings.

When the distributions of training and deployment data differ, it is well known that not only data selection but also data weighting methods based on the theory of covariate shift can be effective. The lack of discussion or comparison regarding such weighting approaches represents a weakness, as it limits the understanding of how the proposed data selection strategy fits within the broader methodological landscape.

Careful consideration is also needed regarding whether the chosen datasets in this study can serve as a representative benchmark for the entire field of domain adaptation. Since experts were involved in both defining the problem and selecting the datasets, there is a potential for bias, and clear explanation is required to account for its possible effects.

**Questions:**

Domain Adaptation has a long research history, with numerous survey papers and performance comparison studies. To more clearly demonstrate the uniqueness and effectiveness of the benchmark dataset constructed in this study, would it be possible to conduct additional experimental evaluations using existing representative benchmark datasets?

When the distributions of training and test (deployment) data differ, density ratio–based weighted learning is considered one of the standard effective approaches. Do the authors have plans to evaluate the performance or behavior of this weighted learning method on the benchmark dataset constructed in this study?

It would be helpful to clarify whether the experts who designed the datasets and problem settings were the same individuals who selected the deployment datasets. Furthermore, if experts were involved in parts of the evaluation experiments, should the paper provide a more detailed explanation of the experimental protocol and the decision-making process to ensure transparency and minimize potential bias?

---

### Meta-Review · Area_Chair_CV4j · 2025-12-16

**Summary:**

No response was given to the weaknesses highlighted by the reviewers. I therefore consider that the authors agree with these criticisms and that they cannot be easily addressed.

**Reviewer Scores:**

N/A

---

### Decision · Program_Chairs · 2026-01-26

Reject